# Hilbert problems for the climate sciences in the 21st century – 20 years later

Michael Ghil[1,2]

[1]Ecole Normale Supérieure and PSL University, Paris, France
[2] University of California at Los Angeles, Los Angeles, California, USA

*Correspondence to:* M. Ghil(ghil@atmos.ucla.edu)

**Abstract.** The scientific problems posed by the Earth's atmosphere, oceans, cryosphere — along with the land surface and biota that interact with them — are central to major socio-economic and political concerns in the 21st century. It is natural, therefore, that a certain impatience should prevail in attempting to solve these problems. The point of a review paper published in this journal in 2001 was that one should proceed with all diligence, but not excessive haste: "festina lente," i.e. "hurry in a measured way." The earlier paper traced the necessary progress through the solutions of ten problems, starting with "What can we predict beyond one week, for how long, and by what methods?" and ending with "Can we achieve enlightened climate control of our planet by the end of the century?"

A unified framework was proposed to deal with these problems in succession, from the shortest to the longest time scale, i.e. from weeks to centuries and millennia. The framework is that of dynamical systems theory, with an emphasis on successive bifurcations and the ergodic theory of nonlinear systems, on the one hand, and on pursuing this approach across a hierarchy of climate models, from the simplest, highly idealized ones, to the most detailed ones. Here, we revisit some of these problems, 20 years later,[1] and extend the framework to coupled climate-economics modeling.

## 1 Introduction and motivation

In order to assess to what extent and in which ways we are modifying our global environment, it is essential to understand how this environment functions. In the past two decades, it has become abundantly clear that we do affect the climate system, both globally and locally (IPCC, 1990, 2001, 2007, 2014a), but many of the uncertainties and missing details are still with us.

We take, therefore, herein a planetary view of the Earth's climate system, of the pieces it contains, and of the way these pieces interact. This will allow us to eventually understand, predict with confidence and with known error margins, and ultimately exert some rational control on the individual pieces, and thus on the whole of such a complex system.

Some readers of the earlier paper will notice a slight change in the title. The climate sciences used in the title now have evolved rather rapidly over the last two decades and have become a fairly broad field in their own right. Rather than casting an even wider net to encompass all of the geosciences, we decided to claim merely the climate sciences as the topic. On the other hand, the problem of mitigating the effects of climate change and adapting to them cannot be solved without a thorough

---

[1]With an obvious nod to "Vingt ans après," the sequel of Alexandre Dumas to his "Three Musketeers."

understanding of basic economic principles. The need for such an understanding and for weaving it into the solution of the last problem has led to the need for casting a wider net in the direction of macroeconomic data analysis and modeling.

Several research groups carried out an important extension of the dynamical-systems and model hierarchy framework of Ghil (2001) during the past two decades, from deterministically autonomous to nonautononomous and random dynamical systems (NDS and RDS: e.g., Ghil et al., 2008; Chekroun et al., 2011; Bódai and Tél, 2012). This framework allows one to deal in a self-consistent way with the increasing role of time-dependent forcing applied to the Earth system by humanity, as well as by natural processes, such as solar variability and volcanic eruptions. Ghil (2019, Sec. 5.3) and Ghil and Lucarini (2020, Sec. IV.E) provided recently a fairly complete review of these advances and we shall thus mention them herein only in passing.

The ten problems proposed in Ghil (2001) to achieve this goal were:

1. What is the coarse-grained structure of low-frequency atmospheric variability, and what is the connection between its episodic and oscillatory description?

2. What can we predict beyond one week, for how long, and by what methods?

3. What are the respective roles of intrinsic ocean variability, coupled ocean-atmosphere modes, and atmospheric forcing in seasonal-to-interannual variability?

4. What are the implications of the answer to the previous problem for climate prediction on this time scale?

5. How does the oceans' thermohaline circulation change on interdecadal and longer time scales, and what is the role of the atmosphere and sea ice in such changes?

6. What is the role of chemical cycles and biological changes in affecting climate on slow time scales, and how are they affected, in turn, by climate variations?

7. Does the answer to the question above give us some trigger points for climate control?

8. What can we learn about these problems from the atmospheres and oceans of other planets and their satellites?

9. Given the answer to the questions so far, what is the role of humans in modifying the climate?

10. Can we achieve enlightened climate control of our planet by the end of the century?

These problems were listed in increasing order of time scale, from the shortest to the longest one, i.e. from weeks to centuries and millennia. Ghil (2001) emphasized the fact that, in mathematics, clearly formulated problems can be given fully satisfactory solutions. Thus, in his "Lecture delivered before the International Congress of Mathematicians at Paris in 1900," David Hilbert[2] proposed ten problems, whose number was increased to 23 in a subsequent publication (Hilbert, 1900). In fact, of the properly formulated Hilbert problems, the ten problems $\{3, 7, 10, 11, 13, 14, 17, 19, 20, 21\}$ have a resolution that

---

[2]The author of this paper is a great-great-grandson of D.H., through the sequence M.G.–P.D. Lax–K.O. Friedrichs–R. Courant–D. H., cf. https://www.genealogy.math.ndsu.nodak.edu/id.php?id=33687, but that is where any similarity or proximity stops.

is accepted by a general consensus of the mathematical community. On the other hand, the solutions proposed for the seven problems $\{1, 2, 5, 9, 15, 18, 22\}$ are only partially accepted as resolving the corresponding problem.

That leaves problems 8 (the Riemann hypothesis), 12 and 16 unresolved, while 4 and 23 were too vaguely formulated to ever be described as solved. Problem 6 is of particular interest to us here. Its overall heading (Hilbert, 1900) is "Mathematical treatment of the axioms of physics," meaning that one should treat them in the same way as the "foundations of geometry." This problem has been interpreted as having two subproblems: (a) an axiomatic treatment of probability that will yield limit theorems for the foundation of statistical physics; and (b) a rigorous theory of limiting processes "which lead from the atomistic view to the laws of motion of continua," e.g. from Boltzmann's equations of statistical mechanics to the partial differential equations of continuous media. The mathematical community considers that the axiomatic formulation of probability theory by Kolmogoroff (1933, reissued in 2019) is an entirely satisfactory solution of part (a), although alternative formulations do exist; part (b) is work in progress.

To the contrary, problems in the physical sciences — let alone in the life sciences or socio-economic sciences — cannot be "solved" in general to everybody's satisfaction in finite time. Apparently, though, social media do entertain the notion of "Hilbert Problems for Social Justice Warriors," whatever that may mean.

The ten original problems of Ghil (2001) could easily be completemented with 13 more and the unanswered problems of the climate sciences would still be far from exhausted. We illustrate instead in the rest of this paper how the attempts to solve four of the ten problems above — Problems 1, 2, 3 and 10 — have fared over the intervening two decades and do so quite succintly. Sections 2 and 3 deal with Problems 1 and 2, and with Problem 3, respectively. Sections 4 and 5, in turn, address two complementary aspects of Problem 10: the climate and coupling part vs. the economic part. Concluding remarks follow in Sec. 6 and Appendix A provides some technical details on the results concerning fluctuation–dissipation in macroeconomics.

## 2 Problems 1 & 2. Low-frequency atmospheric variability and medium-range forecasting

In the climate sciences, like in all the sciences, terms like "low-frequency" and "long-term" have to be defined quantitatively. The dominant frequency band in midlatitude day-to-day weather is the so-called synoptic frequency of the evolution of extra-tropical weather systems, which corresponds to periodicities of 5–10 days. Thus, for the atmosphere, low-frequency variability (LFV) and medium-range forecasting refer to time intervals longer than 10 days.

As recently mentioned in Ghil et al. (2018) and Ghil and Lucarini (2020), it was John von Neumann's (1903–1957), at the very beginnings of climate dynamics, who made an important distinction (Von Neumann, 1960) between weather and climate prediction. To wit, short-term numerical weather prediction (NWP) is the easiest form of prediction — i.e., it is a pure initial-value problem; long-term climate prediction is next easiest — it corresponds to studying the system's asymptotic behavior; while intermediate-term prediction is hardest — both initial and boundary values are important. In this case, the boundary values refer mainly to the boundary conditions at the air–sea and air–land interfaces.

Essentially, the first of the three problems above corresponds to E. N. Lorenz's predictability of the first kind, while the second one corresponds to his predictability of the second kind (Lorenz, 1967; Peixoto and Oort, 1992). It is the intermediate-

term prediction that requires going beyond the initial-value problem but without reaching all the way to a statistical equilibrium for very long times. It is this problem that requires a unified treatment of slower climate change in the presence of faster climate variability and we return to it in Secs. 3 and 4.

Concerning the study of atmospheric LFV and medium-range forecasting, Ghil (2001) had little to say about them at the time. Both areas of inquiry, though, have taken huge strides over the last two or three decades (e.g., Kalnay, 2003; Palmer, 2017): the weather forecast for planning one's holidays at the beach or in the mountains next week has become considerably more reliable. Still, a key issue associated with Problem 1 was formulated by Ghil and Robertson (2002), namely whether it is the "wave" point of view or the "particle" one that is more helpful in observing, describing and predicting LFV. To wit, is it (i) oscillatory modes with periods of 30 days and longer, namely the waves, or (ii) persistent anomalies with durations of 10 days or longer and the Markov chains of transitions between more or less persistent regimes, namely the particles, that are more interesting and useful in coming to grips with medium-range forecasting?

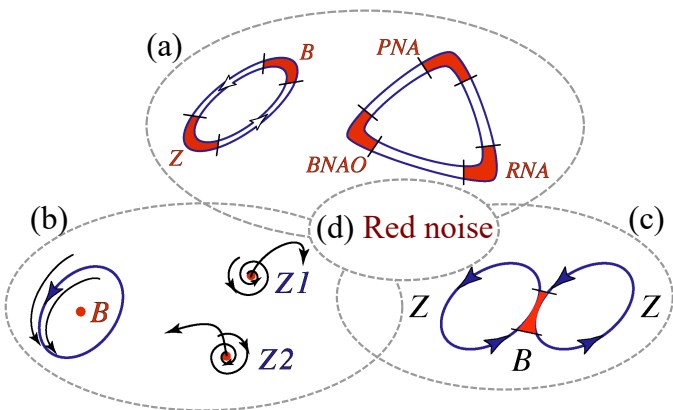

**Figure 1.** Schematic overview of atmospheric LFV mechanisms. Reproduced from Ghil et al. (2018) with permission from Elsevier.

Ghil et al. (2018) have reformulated this problem more completely in Fig. 1. Here the diagram (a) represents Markov chains between two or more flow regimes with distinct spatial patterns and stability properties, such as blocked ($B$) and zonal ($Z$) (Charney and DeVore, 1979, and references therein) or Pacific–North-American ($PNA$), Reverse PNA ($RNA$) and the blocked phase of the North Atlantic Oscillation ($BNAO$) (Kimoto and Ghil, 1993a; Smyth et al., 1999).

Diagram (b) in the figure is associated with the idea of oscillatory instabilities of one or more of the multiple fixed points that can play the role of regime centroids. Thus, Legras and Ghil (1985) found a 40-day oscillation due to a Hopf bifurcation off their blocked regime $B$, while $Z_1$ and $Z_2$ in their model were generalized saddles that both had zonal-flow patterns. An ambiguity arises, though, between this point of view and a complementary possibility, namely that the regimes are just slow phases of such an oscillation, caused itself by the interaction of the mid-latitude jet with topography Thus, Kimoto and Ghil (1993b) found, in their observational data, closed paths within a Markov chain whose states resemble well-known phases of an intraseasonal oscillation. Furthermore, multiple regimes and intraseasonal oscillations can coexist in a two-layer model on the sphere within the scenario of "chaotic itinerancy" (Itoh and Kimoto, 1997).

Diagram (c) in Fig. 1 is a sketch the linear point of view that persistent anomalies in mid-latitude atmospheric flows on 10–100-day time scales are just due to the slowing down of Rossby waves or to their linear interference (Lindzen, 1986). An interesting extension of this approach into the nonlinear realm is due to N. Nakamura and associates (Nakamura and Huang, 2018; Paradise et al., 2019). The traffic jam analogy for blocking in this work is somewhat similar to the hydraulic jump analogy of C. G. Rossby and collaborators (1939); see also Malone et al. (1951/1955/2016, p. 432).

Finally, diagram (d) corresponds to the effects of stochastic perturbations on any of the (a)–(c) scenarios (Hasselmann, 1976; Kondrashov et al., 2006; Palmer and Williams, 2009).

Recently, Lucarini and Gritsun (2020) made an interesting step in reconciling scenarios (a) and (b) in the figure. These authors used a fairly realistic, three-level quasi-geostrophic (QG3) model (Marshall and Molteni, 1993; Kondrashov et al., 2006) to study blocking events through the lens of unstable periodic orbits (UPOs: Cvitanović and Eckhardt, 1989; Gilmore, 1998). UPOs are natural modes of variability that populate densely a chaotic system's attractor. Lucarini and Gritsun (2020) found that blockings occur when the system's trajectory is in the neighborhood of a specific class of UPOs.

The UPOs that correspond to blockings in the QG3 model are more unstable than the UPOs associated with zonal flow; thus blockings are associated with anomalously unstable atmospheric states, as suggested theoretically by Legras and Ghil (1985) and confirmed experimentally in a rotating annulus with bottom topography by Weeks et al. (1997); see also Ghil and Childress (1987/2012, Ch. 6). Different regimes (particles) may be associated with different bundles of UPOs (waves).

Given this perspective on atmospheric LFV, what can be said about the predictability of flow features in the 10–100-day window, between the limit of detailed, deterministic predictability, on the one hand (e.g., Lorenz, 1969), and the large changes induced in the atmospheric circulation by the march of seasons, on the other? Clearly, the occurrence of certain flow patterns that are more frequently observed, and thus associated with clusters or regimes, should be more predictable. The relative success of Markov chains in describing the transitions between qualitatively different regimes is consistent with the results of Lucarini and Gritsun (2020).

Ghil et al. (2018) have carried out a detailed review of many studies on what used to be called intraseasonal atmospheric variability and is being called more recently subseasonal-to-seasonal (S2S) variability. They concluded that the number and variety of methods that have been used to identify and describe LFV regimes are leading up to a tentative consensus on their existence, robustness, and characteristics. S2S forecasting has become operationally viable and is under intensive investigation (e.g., Robertson and Vitart, 2018, and references therein).

## 3  Problem 3. Oceanic interannual variability

A remarkable feature of human nature is tending to always put the blame elsewhere than at one's own doorstep. Thus meteo-rologists tended to blame sea surface temperatures (SSTs) for changes in atmospheric circulation on S2S time scale and longer, while oceanographers blamed changes in the wind stress for such changes in the upper ocean. It is more judicious, though, to ask "What are the respective roles of intrinsic ocean variability, coupled ocean-atmosphere modes, and atmospheric forcing in seasonal-to-interannual variability?" — as done in Problem 3 of Ghil (2001).

The difference between the two approaches is largely one between linear thinking, in which changes in a system at frequencies not present in its free modes have to be due to external agencies, and nonlinear thinking, in which combination tones and other more complex spectral features may be present. Moreover, interactions between subsystems and between any subsystem and time-dependent forcing can be much richer in a nonlinear world. We will briefly sketch here the evolution of the latter point of view in the study of oceanic interannual variability.

A paradigmatic example of how complex intrinsic LFV can arise in the ocean circulation is the so-called double-gyre problem (e.g., Ghil et al., 2008; Ghil, 2017). Note that the synoptic time scale in the oceans is associated with the oceanic counterpart of "weather" — i.e., with the lifetime of so-called mesoscale eddies — and it is of months rather than a week or two (Gill, 1982; Pedlosky, 1996). Hence LFV in the ocean corresponds to several years rather than to one-to-three months.

Veronis (1963) already obtained bistability of steady solutions in a single-gyre configuration, as well as a stable limit cycle for time-independent wind stress. Jiang et al. (1995) studied the successive-bifurcation tree all the way to chaotic solutions in a double-gyre model with steady, time-independent forcing. The periodic solutions they obtained were pluriannual, had the characteristics of relaxation oscillations, and were termed gyre modes because of the strong vortices they exhibited on either side of the separation of the model's eastward jet from the western boundary (Dijkstra and Ghil, 2005).

Pierini et al. (2016, 2018) applied to simplified double-gyre models the previously mentioned NDS theory. These authors found that, even in the presence of time-dependent forcing and of a unique global pullback attractor (PBA), two local PBAs with very different stability properties can coexist and that their mutual boundary appears to be fractal; see Fig. 2 here and the more detailed explanations in Ghil (2019, Fig. 12). Ghil (2017) and Ghil and Lucarini (2020) reviewed both the fundamental ideas of NDS and RDS theory and their applications to climate problems; hence little more will be said herein on these topics.

Feliks et al. (2004, 2007) showed that a narrow and sufficiently strong SST front with the 7-year periodicity of the oceanic gyre modes could give rise to a similar near-periodicity in the atmospheric jet stream above the oceanic eastward jet, provided the resolution of the atmospheric model was sufficiently high; see also Minobe et al. (2008). Groth et al. (2017) studied reanalysis fields for both ocean and atmosphere over the North Atlantic basin and adjacent land areas (25 N–65 N, 80 W–0); they found their results to be in good agreement with the dominant 7-8-year periodicity of the North Atlantic Oscillation (NAO) being due to the barotropic gyre modes' intrinsic periodicity. The agreement with the alternative theory of the turbulent oscillator (Berloff et al., 2007) playing a key role in the NAO was less evident, since the latter depends in an essential way on strong baroclinic activity and has a much broader spectral peak that does not emphasize the NAO's 7-8-yr peak.

On the other hand, Vannitsem et al. (2015) investigated oceanic LFV in a coupled ocean–atmosphere model with a total of 36 Fourier modes. Their results included stable decadal-scale periodic orbits with a strong atmospheric component, as well as chaotic solutions that were still dominated by the decadal behavior. Projecting atmospheric and oceanic reanalysis data sets onto the leading modes of the Vannitsem et al. (2015) model, Vannitsem and Ghil (2017) confirmed that a dominant LFV signal with a 25–30-year period (Timmermann et al., 1998; Frankcombe and Dijkstra, 2011) is a common mode of variability of the atmosphere and oceans.

Clearly, the separation between the wind-driven circulation addressed by Problem 3 and the buoyancy-driven circulation addressed by Problem 5 is rather a matter of convenience: a water particle in the oceans is affected by both types of forces.

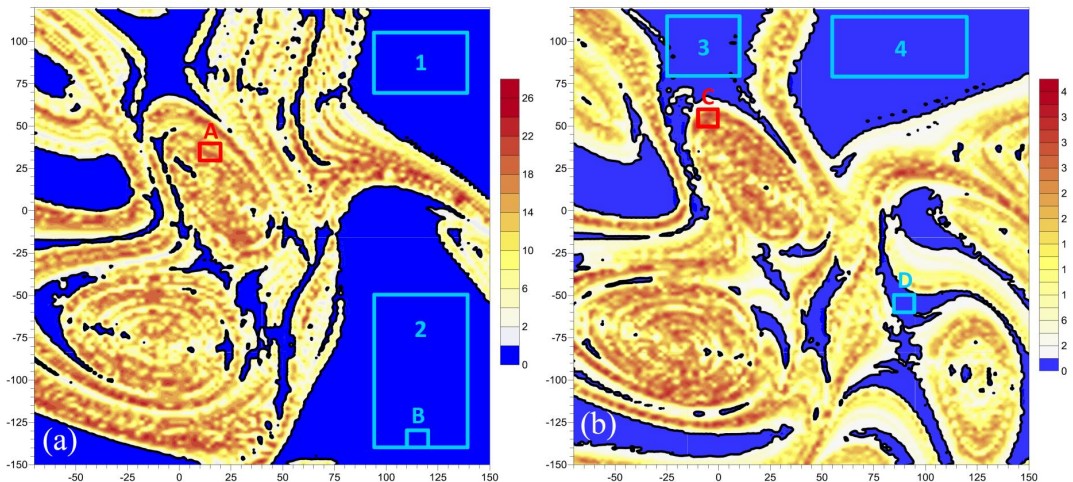

**Figure 2.** Numerical evidence for the existence of two local PBAs in the wind-driven mid-latitude ocean circulation. Plotted is a mean normalized distance $\Delta$ for 15 000 trajectories of the double-gyre ocean model of Pierini et al. (2016, 2018); the cold colors correspond to very quiescent behavior, while the warm colors are associated with unstable, chaotic motion on the PBA. The parameter values in the two panels are, respectively, subcritical and supercritical in the autonomous version of the model with respect to the homoclinic bifurcation that gives rise to relaxation oscillations in the latter: (a) $\gamma = 0.96$ and (b) $\gamma = 1.1$. Reproduced from Pierini et al. (2016). ©American Meteorological Society; used with permission.

Moreover, recently, Cessi (2019, and references therein) have argued that the meridional overturning is actually powered by momentum fluxes and not by buoyancy fluxes. This argument is not quite generally accepted; see, for instance, Tailleux (2010). Given the lack of consensus in the matter, the thermohaline circulation of Problem 5 is termed increasingly the oceans' meridional overturning circulation, thus avoiding a definite attribution of its physical causes.

5    In the studies of atmospheric, oceanic and coupled variability of the climate system, considerable progress has been made in applying dynamical systems theory and, in particular, bifurcation theory to models subject to time-dependent forcing (Alkhayuon et al., 2019) or further towards the high end (Rahmstorf et al., 2005; Hawkins et al., 2011) of the model hierarchy originally proposed by Schneider and Dickinson (1974). More recently, Ghil (2001) and Held (2005), among others, have emphasized the need to pursue such a hierarchy systematically in order to further increased understanding of the climate system and of its predictability, rather than merely push to higher and higher resolution in order to achieve ever more detailed simulations of the system's behavior for a limited set of semi-empirical parameter values..

## 4 Problem 10A. Climate change and its control: Integrated thinking

### 4.1 Background

Much more has been done on this ultimate problem over the last two decades than over the two previous ones. First of all, it has become obvious that we cannot wait till the end of the century to achieve enlightened control over the climate. The attribute "enlightened" here plays a crucial role: it clearly does not include rather crude geoengineering proposals that risk doing as much or more harm than good. The field of geoengineering has blossomed, though, and we merely refer here to a recent critique of some of the more misguided proposals (Bódai et al., 2020); see also Ghil and Lucarini (2020, Sec. IV.E.4).

Some combination of reduction of greenhous gas emissions, increase of capture and sequestration, and a variety of adaptation and mitigation strategies has to be implemented to avoid the most dire consequences of anthropogenic climate change (Stern, 2007; Nordhaus, 2013; IPCC, 2014b). Large uncertainties, however, remain and have to be taken into account in the decision processes leading to near-optimal and affordable strategies, as well as in the implementation thereof.

**Detection and attribution (D&A) studies.** Before addressing these issues, it is worth mentioning that important strides have been taken in the field of detection and attribution (D&A) of individual events to climate change (Stone and Allen, 2005; Hannart et al., 2016b). To start, changes in global quantities that involve averaging over large spans of time and large areas of the globe have been both detected and attributed, with considerable confidence, to anthropogenic changes in the atmospheric concentration of aerosols and greenhouse gases (IPCC, 2014a, b). The D&A of regional changes (e.g., Stott et al., 2010) and, a fortiori, of individual events (e.g., Hannart et al., 2016a) is considerably more difficult and much less incontrovertible.

Given the substantial impact of extreme events on human life and socio-economic well being (e.g., Ghil et al., 2011; Chavez et al., 2015; Lucarini et al., 2016), an important step in achieving greater rigor in this field is a greater reliance on the counterfactual theory of necessary and sufficient causation due to J. Pearl (Pearl, 2009a, b) in the attribution of such events.

The counterfactual definition of causality goes back to the Scottish Enlightenment philosopher, historian, economist, and essayist David Hume (1711–1776), widely remembered for his empiricism and skepticism. It can be stated simply as follows: $Y$ is caused by $X$ if and only if $Y$ would not have occurred were it not for $X$.

The usual identification of the J. Pearl's causal theory as "counterfactual" appears to be, at first sight, rather counterintuitive. We take, therefore, a little detour here to explain briefly the theory, as well as outline how it differs from the usual approach taken so far in D&A studies (Allen, 2003; Stone and Allen, 2005). In doing so, we follow Hannart et al. (2016b).

An individual event is characterized by a binary variable $Y \in \{0, 1\}$, say the threshold exceedance of surface air temperatures for a time interval $\tau$ and over an area $A$. For brevity, we will use the "event $Y$" as a stand-in for the event defined by $\{Y = 1\}$. The idea of causation of $Y$ by a difference $f \in \mathcal{F}$ in the forcing — with $\mathcal{F}$ representing a set of values of insolation, atmospheric composition, etc. — is to distinguish between a situation in which $f$ has the value measured during $Y$ in the real, or factual, world and the value $f = 0$ that it would have had in an alternative, or counterfactual, world. The presence or absence of the extra forcing $f$ is captured by another binary variable, $X_f$.

Obviously, the distinction between the two situations requires one's estimating the probability $p_1 = P(Y = 1 | X_f = 1)$ of the event occurring in the factual world and the probability $p_0 = P(Y = 1 | X_f = 0)$ of its occurring in the counterfactual world.

The prevailing approach is, given estimates $p_1$ and $p_0$, t compute the so-called fraction of attributable risk $F_{\mathrm{AR}}$

$$F_{\mathrm{AR}} = 1 - p_0/p_1. \tag{1}$$

We skip here several important steps in causality theory that involve comparing directed dependency graphs when one is interested in more than one possible effect — e.g., a dust devil and a hailstorm — and more than one cause may be at play, such as the values of the temperature field and those of the wind field in some neighborhood of the observed event. Please see (Hannart et al., 2016b, Fig. 1) and discussion thereof, as well as Pearl (2009b, Sec. 2).

The key mathematical novelty in J. Pearl's counterfactual theory of causation is the realization that, following D. Hume, a cause should be both necessary and sufficient in order to unambiguously atribute an observed event to it. Instead of merely computing the fraction of attributable risk $F_{\mathrm{AR}}$, as in Eq. (1), one needs to define and compute the probabilities PN, PS and PNS of necessary, sufficient, and necessary and sufficient causation.

Thus, the probability $P_{\mathrm{N}}$ of necessary causation is defined as the probability that the event $Y$ would *not* have occurred in the *absence* of the event $X$ given that both events $Y$ and $X$ did *in fact* occur. Sufficient causation, on the other hand, means that $X$ always triggers $Y$ but that $Y$ may also occur for other reasons without requiring $X$. Finally, $P_{\mathrm{NS}}$ is the probability that a cause is both necessary and sufficient. These three definitions are formally expressed as follows:

$$P_{\mathrm{N}} \equiv P(Y_0 = 0 \mid Y = 1, X = 1), \tag{2a}$$

$$P_{\mathrm{S}} \equiv P(Y_1 = 1 \mid Y = 0, X = 0), \tag{2b}$$

$$P_{\mathrm{NS}} \equiv P(Y_0 = 0, Y_1 = 1). \tag{2c}$$

Recall that the subscript 1 refers to the factual world, while the subscript 0 refers to the counterfactual one, cf. Eq. (1).

The definitions in Eq. (2) are precise and unambiguously implementable, as long as a fully specified probabilistic model of the world is formulated. Under certain assumptions, spelled out by Hannart et al. (2016b), the probabilities $P_{\mathrm{N}}$, $P_{\mathrm{S}}$ and $P_{\mathrm{NS}}$ can be calculated as

$$P_{\mathrm{N}} = 1 - \frac{p_0}{p_1}, \quad P_{\mathrm{N}} = 1 - \frac{1-p_1}{1-p_0}, \quad P_{\mathrm{NS}} = p_1 - p_0. \tag{3}$$

One can easily see that $P_{\mathrm{N}}$ is more sensitive to $p_0$ than to $p_1$ and, conversely, that $P_{\mathrm{S}}$ is more sensitive to $p_1$ than to $p_0$: necessary causation is enhanced further by an event being rare in the counterfactual world, whereas sufficient causation is enhanced further by its being frequent in the factual one; see, for instance, Hannart et al. (2016b, Fig. 2).

An interesting idea — first articulated by Hannart et al. (2016b) and further implemented by Carrassi et al. (2017) — is to apply data assimilation methodology (e.g., Bengtsson et al., 1981; Ghil and Malanotte-Rizzoli, 1991; Kalnay, 2003) for the computation of these three probabilities, using observations from the factual world and a model that encapsulates the knowledge of the system's evolution. One uses two versions of the latter model, the factual one with $X_f = 1$, the other with $X_f = 1$, and the data are supposed to tell one whether $P_{\mathrm{NS}}$ is sufficiently close to unity or not.

**Beyond equilibrium climate sensitivity.** Returning now to the issues of near-optimal control of climate change, it is important to realize that what needs to be controlled is not just the global-and-annual mean surface air temperature $\bar{T}$, as originally

studied in the Charney et al. (1979) report. To outline the progress made in the four decades since the Charney report in thinking about anthropogenic effects on climate, please consider Fig. 3 herein.

The figure is a highly simplified, conceptual diagram of the way that anthropogenic changes in radiative forcing would change the behavior of a climate system with increasingly complex characteristics, as one proceeds from Fig. 3(a) through Fig. 3(b) and on to Fig. 3(c). Therefore, neither the time $t$ on the abscissa nor the $CO_2$ concentration and temperature $\bar{T}$ on the ordinate are labeled quantitatively in the three panels. The time we think of is years to decades and the ranges of the $CO_2$ concentration and $\bar{T}$ correspond roughly to those expected for the difference in values between the end of the 21[st] century and the beginning of the 19[th] century. To keep things as simple as possible — but definitely not any simpler — anthropogenic changes in radiative forcing have been represented by a sudden jump in $CO_2$ concentration, as in the Charney report.

The climate model represented in the figure's panel (a) can be as simple as a forced, linear, scalar ordinary differential equation representing an energy balance model

$$\dot{x} = -\lambda(x - H(t)x_1), \quad x(t) \equiv \bar{T}(t) - \bar{T}_0, \tag{4}$$

with $\lambda > 0$ and $H(t)$ a Heaviside function that jumps from $H = 0$ for $t \leq 0$ to $H = 1$ for $t > 0$. Here $\bar{T}_0$ and $\bar{T}_1$ are the model's equilibrium climates for the radiative forcings before and after the jump, respectively, while $\lambda$ gives the rate of exponentially approaching the new equilibrium $\bar{T}_1$. Note that, in Eq. (4), $x_1 = \bar{T}_1 - \bar{T}_0$.

The case of panel (b) in the figure can be thought of as an idealized climate system in which the El Niño–Southern Oscillation (ENSO) would be perfectly periodic, rather than having an irregular, 2–7-year periodicity, with additional periodicities and chaotic components present, as in panel (c). There are no serious doubts as to the long-term mean $\bar{T}_1$ after the jump being larger than the preindustrial or current $\bar{T}_0$. But figuring out the higher moments of the long-term probability density function (pdf) after the jump is another matter entirely.

Recently, increasing attention has been paid by high-end modelers to the difficulties posed by the presence of internal variability in the climate system. For instance, Deser et al. (e.g., 2020, and references therein) point to this variability's imperfect simulation and to its consequences for attempts at predicting future climates on multidecadal time scales.

Concerning ENSO's distribution of extreme events, Ghil and Zaliapin (2015) investigated its dependence in an idealized delay differential equation (DDE) model on several model parameters. They also found that plotting the model's PBA with respect to the seasonally periodic forcing provided a much better understanding of the role of the seasonal cycle in the model.

Chekroun et al. (2018) found that parameter dependence in such a DDE model can lead to a critical transition between two types of chaotic behavior, which differ substantially in their distribution of extreme events. This contrast is clearly apparent in Fig. 4 and it illustrates the types of nonequilibrium climate changes suggested by Fig. 3(c) above.

The changes in the invariant, time-dependent measure $\mu_t$ supported on this ENSO model's PBA are plotted in panels (a)–(c), as a function of the control parameter $a$. The change in the PBA is clearly associated with the population lying towards the ends of the elongated filaments apparent in the figure. This population represents strong warm, El Niño and cold, La Niña events.

The PBA experiences a critical transition at a value $a_*$; here $h(t)$ is the thermocline depth anomaly from seasonal depth values at the domain's eastern boundary, with $t$ in years, $a = (1.12 + \delta))/180$ and $0.015700 < \delta_* < 0.015707$. Thus, $\mu_t(a)$

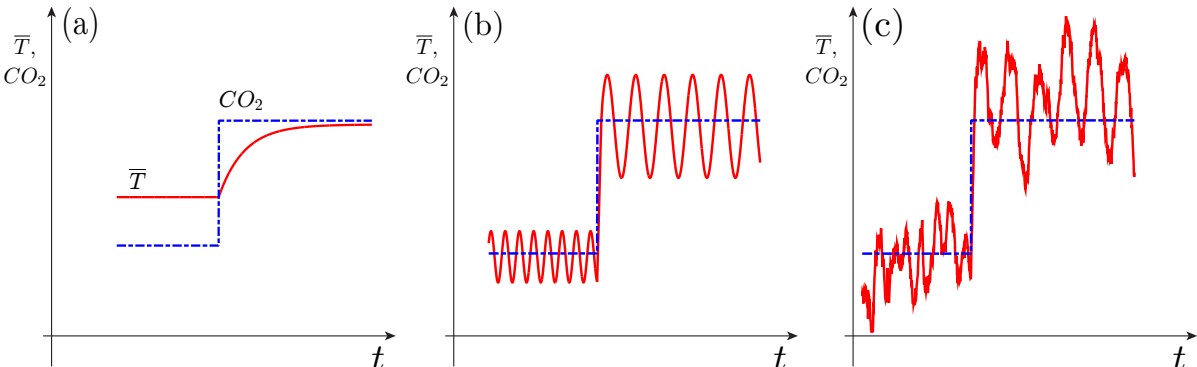

**Figure 3.** Schematic diagram of the effects of a sudden change in atmospheric carbon dioxyde ($CO_2$) concentration (blue dash-dotted line) on seasonally and globally averaged surface air temperature $\bar{T}$ (red solid line). Climate sensitivity (a) for an equilibrium model; (b) for a non-equilibrium, oscillatory model; and (c) for a non-equilibrium, chaotic model, including possibly random perturbations. As a radiative forcing (atmospheric $CO_2$ concentration, say) changes suddenly, global temperature ($\bar{T}$) undergoes a transition: in panel (a) only the mean temperature changes; in panel (b) the mean adjusts, as it does in panel (a), but the period, amplitude and phase of the oscillation can also decrease, increase or stay the same, while in panel (c) the entire intrinsic variability changes as well, including the distribution of extreme events. Reproduced from Ghil (2017), with Open Access under the Creative Commons Attribution License.

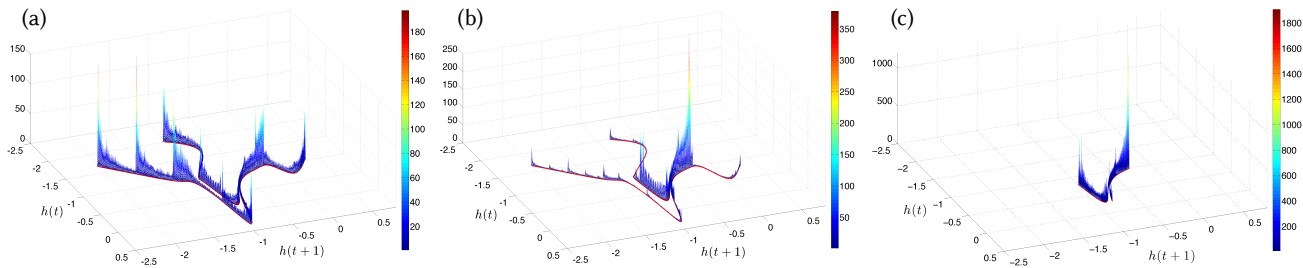

**Figure 4.** Critical transition in extreme-event distribution in an idealized ENSO model: the invariant, time-dependent measure $\mu_t$ supported on the PBA of the ENSO DDE model of Tziperman et al. (1994) is plotted here via its embedding into the $(h(t), h(t+1))$-plane for $a = (1.12 + \delta))/180$, $t \simeq 147.64$ yr and, respectively, (a) $\delta = 0.0$; and (b) $\delta = 0.01500$; and (c) $\delta = 0.015707$. The red curves in the three panels represent the singular support of the measure. After Chekroun et al. (2018), reprinted by permission from Springer International Publishing AG ©2018.

encripts faithfully the disappearance of such extreme events as $a \nearrow a_*$. Adding stochastic perturbations to the model can smooth out the transition, which might make it less drastic in high-end models as well as in observations. Again the study of the model's PBA greatly facilitates the understanding of the processes involved.

## 4.2 Integrated assessment models (IAMs)

So-called integrated assessment models (IAMs) have been so far the main tool to assess the future impact of climate change on the global economy and, even more ambitiously, on one or more regional ones (e.g., Stern, 2007; Nordhaus, 2014; Clarke et al., 2014; IPCC, 2014b; Hughes, 2019). The main purpose of IAMs is to provide reasoned, scientific input into major socio-economic and political decisions that will affect both the present and future generations of humanity, as well as planet Earth as a whole. In doing so, IAMs attempt to weigh the cost and effectiveness of competing or complementary adaptation and mitigation measures by applying various methods of cost–benefit analysis (Clarke et al., 2014; IPCC, 2014b; Hughes, 2019) and decision theory (e.g., Barnett et al., 2020, and references therein).

IAMs attempt to link major features of economy and society with the climate system and biosphere into one modelling framework, a lofty purpose that clearly has to overcome major obstacles. Some of the latter have to do with the complexity of the coupled system's distinct components, others with the different cultures and research styles of the scientific communities involved. Finally, the data sets necessary to estimate model parameters are short, incomplete and often rather inaccurate. The United Nation's Intergovernmental Panel on Climate Change (IPCC) has dedicated substantial efforts over the last three decades to overcoming these various obstacles (IPCC, 1990, 2001, 2007, 2014a, b).

Mostly, IAMs have used both climate and economic modules that were conceived in the spirit of Fig. 3(a), i.e., (i) of so-called equilibrium climate sensitivity (ECS), as studied by Charney et al. (1979) four decades ago, for the climate module; and (ii) of general equilibrium theory (Walras, 1874/1954; Pareto, 1919; Arrow and Debreu, 1954) going back to the late 19th century, for the economic module. We have considered in Secs. 2 and 3 above how to formulate a more active climate module that might behave more like Figs. 3(b) or even 3(c), given changes in radiative forcing induced by anthropogenic emissions of greenhouse gases and aerosols. For illustration purposes, we will merely sketch a highly idealized counterpart of such behavior for the economic module of an IAM.

General equilibrium theory is a cornerstone of today's mainstream economics, often referred to as neoclassical (Aspromourgos, 1986). This theory relies heavily on equilibrium in both the labor and product markets: prices of goods and wages of labor are assumed to be flexible and to adjust so as to achieve equilibrium in the product and labor markets at all times. As a result, it is possible to maximize an intergenerational utility functional, following the planning approach of Ramsey (1928). Moreover, the mean growth of the economy (Solow, 1956) is only perturbed by exogenous shocks that lead to random fluctuations reverting to a stable equilibrium, such as can be modeled by auto-regressive processes of order 1, called AR(1) processes.

The economic modules of most IAMs used so far in the IPCC process (e.g., IPCC, 2014b, and references therein) rely on general equilibrium theory and its consequences. These IAMs differ largely by the values they prescribe for various parameters; among the latter, the most important one is the discount factor, which essentially gives the future value of a currency unit vs. its value today. Large differences among the value of this factor assumed in the work of Stern (2007) vs. that of Nordhaus (2014), for instance, have lead to very different conclusions about mitigation policies recommended by these two authors.

More generally, Wagner and Weitzman (2015, among others) have emphasized how uncertainty in the climate system's dynamics could create fat-tailed distributions of potential damages, while Pindyck (2013) and Morgan et al. (2017) find existing

IAMs to be of little value in guiding prudent adaptation and mitigation policy. More radically, Davidson (1991) already questioned the extent to which certain types of economic uncertainties could be represented judiciously by probabilistic approaches, as done routinely in the IAMs' estimation of utility functionals associated with the system's future trajectories. Farmer et al. (2015) have also emphasized the need for better uncertainty estimates, better accounting for technological change and for heterogeneities in the coupled system, as well as for more realistic damage functions.

More specifically, Barnett et al. (2020) have recently emphasized that the uncertainties associated with assessing the future impact of climate change, and hence with devising adaptation and mitigation policies, go well beyond the well-known uncertainties in the discount factor and in other parameters of either the climate or the economic module of coupled models. They suggest three much broader types of uncertainties:

(i) *risk* – uncertainty within a model: uncertain outcomes with known probabilities;

(i) *ambiguity* – uncertainty across models: unknown weights for alternative possible models; and

(i) *misspecification* – uncertainty about models: unknown flaws of approximating models.

It is worth considering, in this context, uncertainties associated with the economic counterpart of natural or intrinsic variability in the climate system; such variability is called endogenous in the economic literature. Following a parallel line of reasoning, Hallegatte (2005) has argued for closed-loop climate–economy modeling, i.e., a two-way feedback interaction that also accounts for multiple time scales in both modules. We turn therewith to the economic part of the modeling and data analysis, as highly pertinent to a truly integrated way of thinking about the Earth system, including the humans that affect it more and more, whatever the exact time at which the Anthropocene (Crutzen, 2006) might have started (Lewis and Maslin, 2015).

## 5 Problem 10B. Nonequilibrium economics, fluctuation–dissipation, and synchronization

### 5.1 Nonequilibrium economic models and a vulnerability paradox

There is no denying that, superimposed on overall global growth in economic activity, are ups and downs well known as recessions and upswings. These shorter-term variations may appear only as small wiggles on a long-term exponential tendency of economic indicators like gross domestic product (GDP) but they are quite severe in the individual experience of households, firms, countries and even the world as a whole. There are two rather distinct approaches to modeling these so-called business cycles: "real" business cycle (RBC) theory and endogenous business cycle (EnBC) theory. The "real" in RBC theory refers to the fact that the theory explains macroeconomic fluctuations as the result of real productivity shocks and does not emphasize monetary or financial aspects of the economy. A good starting point for this literature is Brock and Mirman (1972).

RBC theory is closely tied to the mainstream economics approach (Kydland and Prescott, 1982), in which the expectations of households and firms are rational, supply equals demand, and there is no involuntary unemployment. In RBC models, the fluctuations are entirely due to external, exogenous shocks and the models' response to such shocks is purely via AR(1) processes. This theory is adopted by a very large fraction of practicing economists, and many modifications to it have tried to

bring it in closer agreement with the observed behavior of real economies (e.g. Hoover, 1992). One way this approach has been criticized is that it describes the world as it ought to be, rather than how it is, and considerable controversy still exists as to its explaining major aspects of observed macroeconomic fluctuations (e.g., Summers, 1997; Romer, 2011).

In contradistinction, EnBC theory relies on a number of heterodox — i.e., non-conformist — economic ideas, most importantly on post-Keynesian economics (Kalecki, 1935; Keynes, 1936/2018; Malinvaud, 1977). EnBC theory acknowledges up front at least some of the imperfections of real economies; in this theory, economic fluctuations are due to intrinsic processes that endogenously destabilize the economic system (Kalecki, 1935; Samuelson, 1939; Flaschel et al., 1997; Chiarella et al., 2005). Even F. A. Hayek, a leading liberal, anti-Keynesian economist, had interesting disequilibrium ideas on the delay between decision and implementation time in investments (Hayek, 1941/2007).

At this point, it might be worth noting that, in equilibrium macroeconomic models, output is supply driven, while in nonequilibrium models it is demand driven, a feature that is inherited by the corresponding models that attempt to assess climate damage. An interesting recent example of the latter is the post-Keynesian DEFINE (Dynamic Ecosystem-FINance-Economy) model, which explicitly includes banks, in addition to firms and households (Dafermos et al., 2018).

*The Nonequilibrium Dynamic Model of Hallegatte et al. (2008).* We present here concisely one particular EnBC model and the role that active economic dynamics may have in modifying the effect of natural hazards on such an economy (Hallegatte and Ghil, 2008). The non-equilibrium dynamical model (NEDyM) of Hallegatte et al. (2008) is a neoclassical model based on the Solow (1956) model, in which equilibrium constraints associated with goods and labor markets clearing are replaced by dynamic relationships that involve adjustment delays. The model has eight state variables — which include production, capital, number of workers employed, wages and prices — and the evolution of these variables is modeled by a set of ordinary differential equations. For a brief summary of the model equations, please see Groth et al. (2015a, Appendix A); the parameters and their values are listed in Hallegatte et al. (2008, Table 3).

NEDyM's main control parameter is the investment flexibility $\alpha_{\mathrm{inv}}$, which measures the adjustment speed of investments in response to profitability signals. This parameter describes how rapidly investment can react to a profitability signal: if $\alpha_{\mathrm{inv}}$ is very large, investment soars when profits are high and collapses when profits are small, while a small $\alpha_{\mathrm{inv}}$ entails a much slower adjustment of the investment to the size of the profits. Introducing this parameter is equivalent to allocating an investment adjustment cost, as proposed by Kydland and Prescott (1982) and by Kimball (1995); these authors found that introducing adjustment costs and delays helps in matching key features of macroeconomic models to the data.

In NEDyM, for small $\alpha_{\mathrm{inv}}$, i.e. slow adjustment, the model has a stable equilibrium, which was calibrated to the economic state of the European Union (EU-15) in 2001 (Eurostat, 2002). As the adjustment flexibility increases, this equilibrium loses its stability and undergoes a Hopf bifurcation, after which the model exhibits a stable periodic solution (Hallegatte et al., 2008).

Business cycles in NEDyM originate from the instability of the profit–investment feedback, which is quite similar to the Keynesian accelerator-multiplier effect. Furthermore, the cycles are constrained and limited in amplitude by the interplay of three processes: (i) A reserve army of labor effect, namely labor costs increasing when the employment rate is high; (ii) the inertia of production capacity; and (iii) the consequent inflation in goods prices when demand increases too rapidly. The model's bifurcation diagram is shown in Fig. 5.

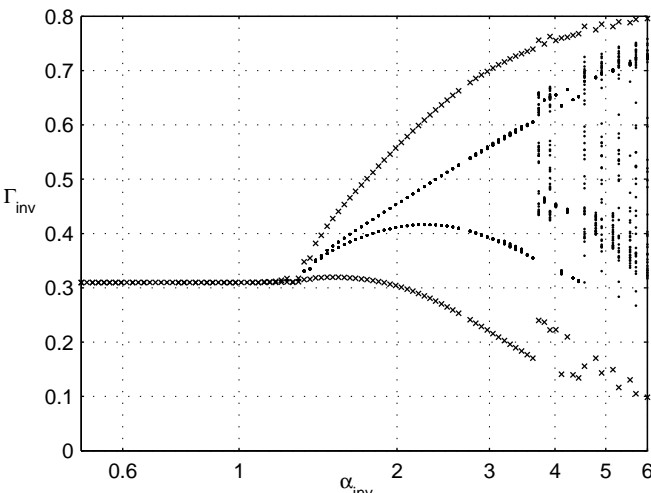

**Figure 5.** Bifurcation diagram of NEDyM, showing its transitions from equilibrium to purely periodic and on to chaotic behavior. The investment parameter $\alpha_{\mathrm{inv}}$ is on the abscissa and the investment ratio $\Gamma_{\mathrm{inv}}$ on the ordinate. The model has a unique, stable equilibrium for low values of $\alpha_{\mathrm{inv}}$, with $\Gamma_{\mathrm{inv}} \simeq 0.3$. A Hopf bifurcation occurs at $\alpha_{\mathrm{inv}} \simeq 1.39$, leading to a limit cycle, followed by transition to chaos at $\alpha_{\mathrm{inv}} \simeq 3.8$. The 'x' symbols indicate first the stable equilibrium and then the orbit's minima and maxima, while dots indicate the Poincaré intersections with the hyperplane $H = 0$, when the goods inventory $H$ vanishes. Reproduced from Groth et al. (2015a) with the permission of AGU Wiley.

For somewhat greater investment flexibility, the model exhibits chaotic behavior, because a new constraint intervenes, namely limited investment capacity. In this chaotic regime, the cycles become quite irregular, with sharper recessions and recoveries of variable duration. In the present paper, we concentrate, for the sake of simplicity, on model behavior in the purely periodic regime, i.e. we have regular EnBCs, but no chaos. Such periodic behavior is illustrated in Fig. 6.

5    The NEDyM business cycle is consistent with many stylized facts described in the macroeconomic literature, such as the phasing of the distinct economic variables along the cycle, called co-movements in the economic literature. The model also reproduces the observed asymmetry of the cycle, with recessions that are much shorter than the expansions. This typical sawtooth shape of a business cycle is not well captured by RBC models, whose linear, auto-regressive character gives intrinsically symmetric behavior around the equilibrium. The amplitude of the price-wage oscillation, however, is too large in NEDyM, 10   calling for a better calibration of the parameters and further refinements of the model.

In the setting of the 2008 economic and financial crisis, the banks' and other financial institutions' large losses clearly reduced access to credit; such a reduction obviously affects very strongly investment flexibility. The EnBC model can thus help explain how changes in $\alpha_{\mathrm{inv}}$ can seriously perturb the behavior of the entire economic system, by either increasing or decreasing the variability in macroeconomic variables, cf. Fig. 5. Moreover, these losses also lead to a reduction in aggregated 15   demand that, in turn, can lead to a reduction in economic production and a full-scale recession.

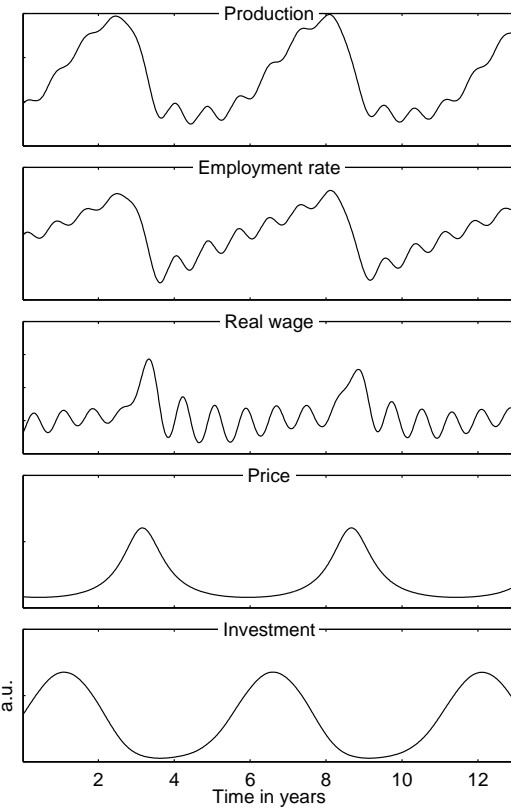

**Figure 6.** Endogenous limit cycle behavior of NEDyM for an investment flexibility of $\alpha_{\text{inv}} = 2.5$; for all other parameter values please see Hallegatte et al. (2008, Table 3). Reproduced from Groth et al. (2015a) with the permission of AGU Wiley.

*Regime-dependent effect of climate shocks.* The immediate damage caused by a natural disaster is typically augmented by the cost of reconstruction, a major concern when considering the disaster's socio-economic consequences. Reconstruction may also lead, though, to an increase in productivity, by allowing for technical changes to be included in the reconstructed capital; technical changes can also sustain the demand and help economic recovery. Economic productivity may be reduced, however,
5  during reconstruction because some vital sectors are not functional, and reconstruction investments crowd out investment into new production capacity (e.g., Hallegatte, 2016, and references therein).

In particular, Benson and Clay (2004, among others) have suggested that the overall cost of a natural disaster might depend on the preexisting economic situation. For instance, the Marmara earthquake in 1999 caused destructions that amounted to 1.5–3 % of Turkey's GDP; its cost in terms of production loss, however, is believed to have been fairly modest due to the fact
10  that the country was experiencing a strong recession of $-7$ % of GDP in the year before the disaster (World Bank, 2003).

To study how the state of the economy may influence the consequences of natural disasters, Hallegatte and Ghil (2008) introduced into NEDyM the disaster-modeling scheme of Hallegatte et al. (2007), in which natural disasters destroy the pro-

ductive capital through a modified production function. Furthermore, to account for market frictions and constraints in the reconstruction process, the reconstruction expenditures are limited.

These authors showed that the transition from an equilibrium to a non-equilibrium regime can radically change the long-term response to exogenous shocks in an EnBC model. Idealized as it may be, NEDyM shows that the long-term effects of a sequence of extreme events depend upon the economy's behavior: a stable-equilibrium economy with very little flexibility or none ($\alpha_{\mathrm{inv}} \lesssim 0.5$, cf. Fig. 5) is more vulnerable than a more flexible economy, albeit still at or near equilibrium (e.g., $\alpha_{\mathrm{inv}} \simeq 1.0$). Clearly, if investment flexibility is null or very low, the economy is incapable of responding to the natural disasters through investment increases aimed at reconstruction; total production losses, therefore, are quite large. Such an economy behaves according to a pure Solow (1956) growth model, where the savings, and therefore the investment, ratio is constant; see Hallegatte and Ghil (2008, Table 1).

When investment can respond to profitability signals without destabilizing the economy, i.e. when $\alpha_{\mathrm{inv}}$ is nonzero but still lower than the critical bifurcation value of $\alpha_{\mathrm{inv}} \simeq 1.39$, the economy has greater freedom to improve its overall state and thus respond to productive capital influx. Such an economy is much more resilient to disasters, because it can adjust its level of investment in the disaster's aftermath.

If investment flexibility $\alpha_{\mathrm{inv}}$ is larger than its Hopf bifurcation value, the economy undergoes periodic EnBCs and, along such a cycle, NEDyM passes through phases that differ in their stability. This in turn leads to a phase-dependent response to exogenous shocks and consequently to a phase-dependent vulnerability of the economic system, as illustrated in Fig. 7.

*The vulnerability paradox.* A key point we wish to make in our excursion into the economical aspects of Problem 10 is precisely this phase dependency of economic response to natural hazards.

In fact, Hallegatte and Ghil (2008) found an interesting vulnerability paradox: the indirect costs caused by extreme events during a growth phase of the economy are much higher than those that occur during a deep recession. Figure 7 illustrates this paradox, by showing in panels (a) a typical business cycle and in panel (b) the corresponding losses for disasters hitting the economy in different phases of this cycle. The vertical lines in both panels, blue at the end of the recession and red in the expansion phase, highlight the paradox. Hallegatte (2016, Sec. 2.2) discussed further aspects of this paradox, and analogous considerations found in the much earlier work of Keynes (1936/2018).

Once noted in NEDyM behavior, this apparent paradox can be easily explained: disasters during high-growth periods enhance pre-existing disequilibria. Inventories are low and cannot compensate the reduced production; employment is high, and hiring more employees induces wage inflation; while the producer lacks financial resources to increase investment. The opposite holds during recessions, as mobilizing investment and labor is much easier (e.g., West and Lenze, 1994).

As a consequence, production losses due to disasters that occur during expansion phases are strongly amplified, while they are reduced when the shocks occur during the recession phase. On average, however, (i) expansions last much longer than recessions, in our NEDyM model as well as in reality; and (ii) amplification effects are larger than damping effects. It follows that the net effect of the cycle is strongly unfavorable to the economy, with an average production loss that is almost as large, for $\alpha_{\mathrm{inv}} = 2.5$, as for $\alpha_{\mathrm{inv}} = 0$; see again Hallegatte and Ghil (2008, Table 1).

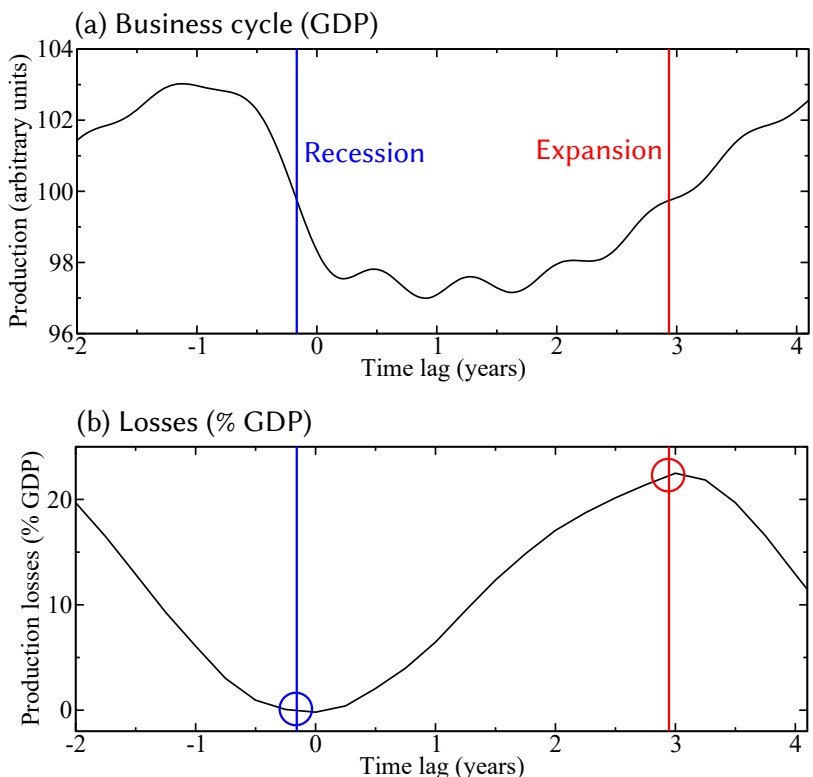

**Figure 7.** Vulnerability paradox: the effect of a single natural disaster on an endogenous business cycle (EnBC). (a) The business cycle in terms of annual production, as a function of time, starting at the cycle minimum (time lag = 0). (b) Total production losses due to a disaster that instantaneously destroys 3 % of gross domestic product (GDP), shown as a function of the cycle phase in which the disaster occurs; phase measured as time lag with respect to cycle minimum. A disaster occurring near the cycle's minimum (blue vertical line in both panels) causes a limited indirect production loss (blue circle), while a disaster occurring during the expansion (red vertical line in both panels) leads to a much larger loss (concentric red circles). Figure courtesy of Andreas Groth, based on the numerical results of Hallegatte and Ghil (2008) and of Groth et al. (2015a).

## 5.2 Fluctuation–dissipation theory (FDT) and synchronization in the economic system

*The fluctuation–dissipation conjecture.* Beyond the obvious implications for disaster assessment, insurance and other practical issues treated by Hallegatte (2016, and references therein), the findings shown schematically in Fig. 7 suggest a theoretically intriguing connection with fluctuation–dissipation theory (FDT) in statistical mechanics. The FDT has its roots in the classical

theory of many-particle systems in thermodynamic equilibrium. The idea goes back to Einstein (1905) and it is very simple: the system's return to equilibrium will be the same whether the perturbation that modified its state is due to a small external impulse or to an internal, random fluctuation. The FDT thus relates natural and forced fluctuations of a system (e.g., Kubo, 1966); it is a cornerstone of statistical physics and has applications in many other areas (Marconi et al., 2008, and references therein). Ghil (2019) and Ghil and Lucarini (2020) have recently reviewed FDT applications in the climate sciences, in the classical form

used for systems in equilibrium (Leith, 1975; Gritsun and Branstator, 2007), as well as in its more recent extensions to systems out of equilibrium based on Ruelle response theory (Ruelle, 1998, 2009; Lucarini, 2008; Lucarini and Gritsun, 2020).

The results in Sec. 5.1 above strongly suggest that the response to exogenous shocks of an economic system might differ from one phase of a business cycle to another. Hence, it is quite possible that the system's endogenous variability might also vary with the phase of a cycle that the system is in. More explicitly, the system's internal, endogenous fluctuations may

change in variance as the phase of the business cycle evolves in the same way as the exogenously driven ones do, i.e., larger "economic volatility" can be expected during expansions than during contractions of the economy. And, if this is the case, out-of-equilibrium response theory (Ruelle, 1998, 2009) may apply to economic systems in the way that it has been found to apply to the climate system, with both local-in-time sensitivity and volatility being phase dependent.

There is a long tradition of systematically analyzing cyclic behavior in economic data (Juglar, 1862; Kitchin, 1923; Burns

and Mitchell, 1946). Yet there is no trace, as far as we could tell, of an investigation along the lines proposed herein. Hence, Groth et al. (2015a, b) set out to study the U.S. macroeconomic data of the Bureau of Economic Analysis (BEA) for 1954–2005 to evaluate the evidence for the FDT conjecture suggested by the results reviewed in Sec. 5.1 above. The nine macroeconomic indicators these authors used were GDP, investment, consumption, employment rate (in %), total wage, change in private inventories, price, exports, and imports; see http://www.bea.gov/.

The nine indicators were each separately detrended by a Hodrick and Prescott (1997) filter, normalized by the trend values, and then collectively analyzed by using a data-adaptive multichannel singular-spectrum analysis (M-SSA) filter; see Ghil et al. (2002), Alessio (2015, Ch. 12) and Groth et al. (2015a, Appendix B) for details. The statistical significance of the results was carefully tested against an AR(1) null hypothesis (Allen and Smith, 1996; Ghil et al., 2002) and they are illustrated in Fig. 8.

The nine detrended and normalized time series are shown in panel (a), with the leading-mode pair of the joint M-SSA

analysis in panel (b). A simple counting of maxima and minima in panel (b) gives 10.5 cycles in 52 years, which agrees rather well with the NEDyM model's 5–6-year period, as well as with the National Bureau of Economic Research's (NBER's) count of 11 cycles for the 65-year interval of 1945–2009, which yields an average period of 69 months = 5.75 years[3].

---

[3]https://www.nber.org/cycles/cyclesmain.html, consulted on April 6, 2020; pdf version dated April 23, 2012

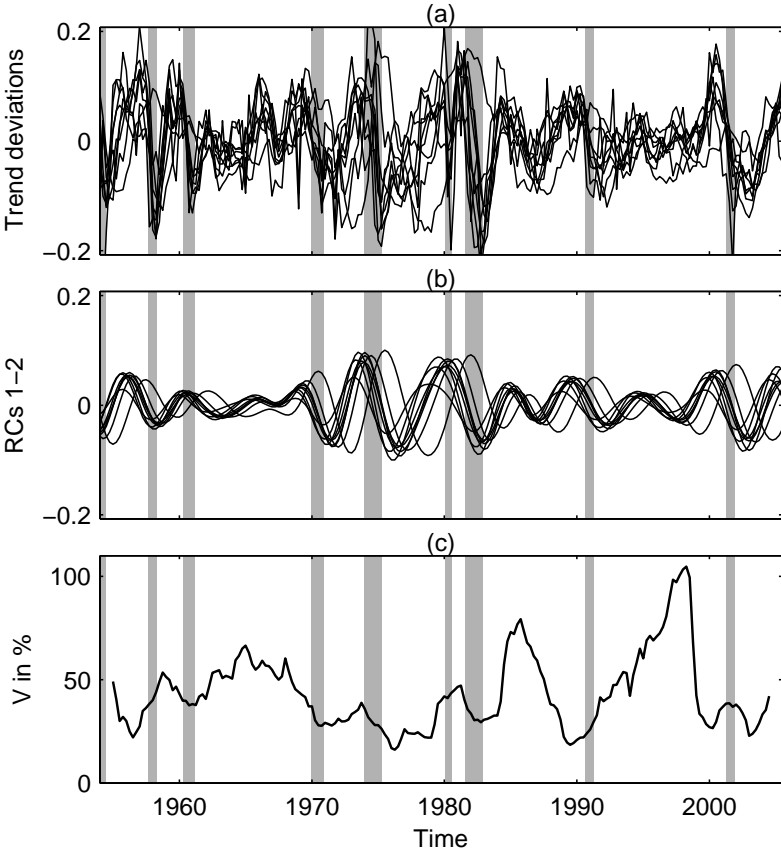

**Figure 8.** US business cycles and the implied fluctuation–dissipation result. Time series of nine U.S. macroeconomic indicators, 1954–2005: (a) Normalized trend residuals; (b) data-adaptively filtered business cycles, captured by the leading oscillatory mode of M-SSA; and (c) local variance $V_{\mathcal{K}}(t)$ of the fluctuations. The shaded vertical bars indicate the NBER-defined recessions, cf. https://www.nber.org/cycles/cyclesmain.html. Reproduced from Groth et al. (2015a) with the permission of AGU Wiley.

In Fig. 8(c) is plotted the evolution in time of the local variance associated with all nine indicators, as measured over a sliding window of $M = 24$ quarters = 6 years. It is clear that the local variance of the fluctuations, as defined in Appendix A, is consistent with the FDT hypothesis, especially over the latter part of the BEA dataset; e.g., the local variance $V_{\mathcal{K}}(t)$ of the fluctuations during the NBER-defined recessions of July 1981 (16 months), July 1990 (8 months), and March 2001 (8 months)

5  is at or very near to a minimum, while substantial local maxima of $V_{\mathcal{K}}(t)$ are attained during the expansions in between.

*Synchronization of economic activity.* Synchronization, known in the 1970s and 1980s as entrainment (Winfree, 1980/2001; Ghil and Childress, 1987/2012), is a key feature of nonlinear oscillators that has been known since Christiaan Huygens' experiment of 1665 in which two pendulum clocks with slightly different lengths synchronized. More recently, the synchronization of chaotic oscillators has become a topic of growing interest in the physical and biological sciences (e.g., Rosenblum et al.,

10  1996; Boccaletti et al., 2002; Pikovsky et al., 2003).

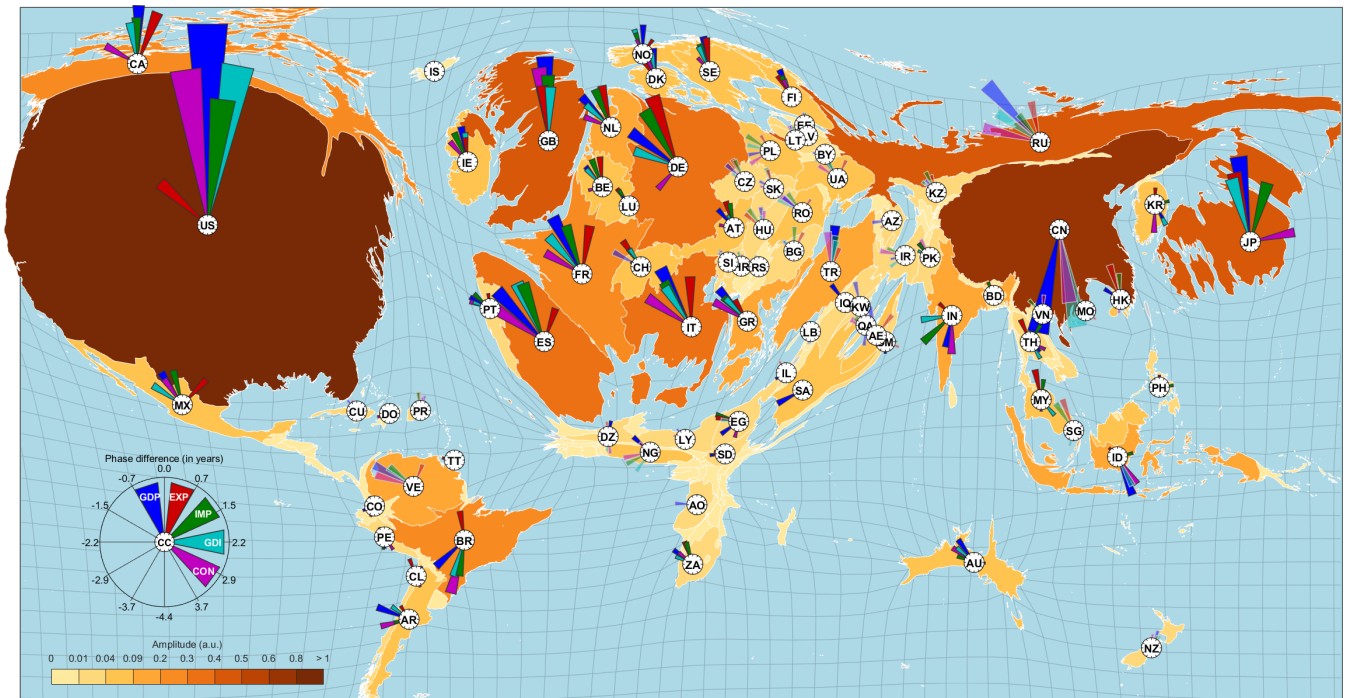

**Figure 9.** Leading mode of synchronized economic activity: world map of this mode's phase and amplitude relations. For each country, the relations among the variables' phase and amplitude are shown in a polar coordinate system, with the two-letter country code at the origin. The corresponding variable codes and colors of the pointers are given in the small compass inset at the lower left. Estimates for variables with missing values are indicated by transparent pointers. Phase differences are given with respect to US GDP in a clockwise manner; i.e., positive and negative values indicate a phase that leads or lags the US GDP, respectively. The land area of each country is proportional to its maximum amplitude over all of its variables. Reproduced from Groth and Ghil (2017) with the permission of AIP Publishing.

Still, while the emergence of business cycle synchronization across countries has been widely acknowledged (Artis and Zhang, 1997; Süssmuth, 2002; Kose et al., 2003) — especially in view of the ongoing globalization of economic activity — no agreement has emerged so far on basic issues like the quantification of comovements. Given the relative shortness of macroeconomic time series, efforts at applying advanced univariate analysis methods to them (e.g, de Carvalho et al., 2012; Sella et al., 2016) have provided interesting but not quite conclusive results.

To overcome the difficulties posed by the simultaneous analysis of a large number of time series, Groth and Ghil (2017) applied a suitable modification of M-SSA (Groth and Ghil, 2011, 2015) to macroeconomic data from the World Development Indicators (WDI) database of the World Bank at http://databank.worldbank.org. The dataset extracted from the WDI database comprised five macroeconomic indicators for 104 countries: GDP, gross fixed capital formation (GDI, formerly gross domestic fixed investment), final consumption expenditure (CON), exports (EXP), and imports (IMP) of goods and services, with all variables in constant 2010 US\$. The data were only analyzed for the 46-year interval 1970–2015, for which at least one of the indicators chosen was available for each of the 104 economies selected. The main result of this analysis is shown in Fig. 9.

The leading mode in the figure has a rough periodicity of 7–11 years, it captures 73 % of the trend residual's variance, and it is statistically significant according to the stringent tests of Groth and Ghil (2011, 2015, 2017). A key ingredient in the tests applied is the much larger number of data points used by the improved M-SSA methodology in examining not just GDP but a more complete set of indicators, with nine time series for Fig. 8 and five for Fig. 9.

The latter figure clearly illustrates the dominance of the U.S. economy over the 1970–2015 time interval, with the UK perfectly aligned on the U.S. indicators, while other European countries, including even the Russian Federation, lag somewhat behind. Japan is also in very good alignment with the US, while China is in almost perfect phase opposition with it, while India and Indonesia follow the Chinese lead. More complicated lead-and-lag patterns are present in the much smaller economies of South America and Africa.

## 6   Concluding remarks

In this review, we have covered purely climate science problems in Secs. 2 and 3, while Secs. 4 and 5 were dedicated, respectively, to coupled climate-economy and purely economic problems. Section 2 dealt with Problems 1 and 2 of Ghil (2001) and it showed progress over the last two decades in bringing more advanced nonlinear methods to bear on the issues of atmospheric low-frequency variability (LFV), as well as progress in extended-range forecasting, especially in the subseasonal-to-seasonal (S2S) range. As clearly stated by Ghil (2001) and again in Sec. 1 herein, physical-sciences problems are less likely than the purely mathematical ones formulated by Hilbert (1900) to be solved to everybody's satisfaction: Fig. 1 still shows a rather broad lack of consensus on the ultimate causes of atmospheric LFV.

In Sec. 3, we examined recent progress in the study of the oceans' wind-driven circulation. A key theme was studying the causes of interannual variability in the midlatitude double-gyre problem and its effect on the atmosphere above. One line of investigation dealt with providing substantial modeling and observational support to the idea that intrinsic ocean variability can have major effects on interannual atmospheric variability, such as the North Atlantic Oscillation.

Another major point touched upon in this section was the use of the theory of nonautononomous and random dynamical systems (NDS and RDS) to treat, in a fully self-consistent way, time-dependent and possibly random forcing by the atmosphere of a dynamically active ocean. A noteworthy finding here is the possibility of multiple modes of behavior, both quiescent and chaotic, for a given set of parameter values, cf. Fig. 2. Finally, a 25–30-year mode of a truly coupled ocean–atmosphere model was discussed,and documented in both models and observations.

In Sec. 4.1, we emphasized the efforts made over the last three decades to gain greater insight into the way that climate change will affect the life of humanity on Earth and, in particular, the world economy. Figure 3 emphasizes that change in climate bears not only on the mean temperatures, but also on the climate system's modes of variability and on the distribution of extreme events. Using here RDS theory is of the essence and has shown already that critical transitions between large and frequent El Niño events and much smaller ones are possible, cf. Fig. 4.

In Sec. 4.2, a quick introduction to integrated assessment models (IAMs) was provided, while emphasizing the equilibrium-based approach in both their climate and economic modules. The very high sensitivity to parameter values of this type of IAMs

has led to rather contradictory results and, therewith, to quite opposite policy recommendations. Section 4 ends with suggesting a broader view of uncertainties than considered heretofore in studying climate change impacts on the world economy.

Section 5 addressed economic aspects of Problem 10, while emphasizing nonequilibrium approaches. We first presented for the geoscientific readership the difference between the real business cycle (RBC) approach, based on general equilibrium theory, and the endogenous business cycle (EnBC) approach, which acknowledges the possibility of imperfect expectations and of the goods and labor markets not clearing, as well as the existence and persistence of involuntary unemployment.

In the latter spirit, we introduced in Sec. 5.1 a highly idealized Nonequilibrium Dynamic Model (NEDyM) and showed, in Fig. 5, its bifurcation sequence, first from equilibrium to purely periodic EnBCs and on to chaotic behavior. In particular, the model exhibits relaxation oscillations with realistically fast contractions and slow expansions; e.g., the mean duration of the U.S. economy's contractions for the post-WWII interval 1945–2009, with 11 cycles, was of 11.1 months, while expansions lasted on average 58.4 months[4]. Such sawtooth behavior cannot be captured by RBC models, in which shocks regress to the mean in AR(1) fashion, independently of the sign of the shock.

The existence of endogenous variability gives rise to a vulnerability paradox, illustrated in Fig. 7, with an exogenous shock producing higher losses during an expansion than during a recession. This asymmetry in response, when integrated over several cycles, produces a net effect that differs from that shown by IAMs based of general equilibrium theory and the so-called Ramsey (1928) planners deduced from that theory.

The vulnerability paradox of Fig. 7 thus led us to the FDT conjecture and its very careful but still tentative verification with U.S. macroeconomic data in Fig. 8. NEDyM, though, is but a highly idealized, aggregate macroeconomic model. It would, therefore, be highly desirable to see such FDT results be reproduced in much more detailed, agent-based models (e.g., Epstein and Axtell, 1996; Bouchaud, 2013; Mazzoli et al., 2019, and references therein). Likewise, producing figures along the lines of Fig. 8 herein with other methods and other datasets would help invalidate, cf. Popper (2005), the present conclusions or, to the contrary, show some consistency with them.

In Sec. 5.2, we turned to the fluctuation–dissipation conjecture suggested by the above vulnerability paradox. To wit, internal, endogenous fluctuations are likely to change in variance with the phase of the business cycle in the same way as the exogenously driven ones, i.e., larger volatility can be expected during expansions than during contractions of the economy. This conjecture was clearly confirmed by the results reproduced in Fig. 8 of an investigation into U.S. macroeconomic indicators. Consequently, out-of-equilibrium response theory (Ruelle, 1998, 2009; Lucarini, 2008) may apply to economic systems in the way that it has been found to apply to the climate system, with both local-in-time sensitivity and volatility being phase dependent

The FDT result captured by Figs. 7 and 8 holds, therewith, great promise for the study of an economic entity's sensitivity to environmental, as well as to economic, political or financial shocks. In general, the usefulness of such a result (Kubo, 1966; Leith, 1975) lies in the fact that one has a much longer record of internal fluctuations than of responses to shocks; see also Ghil (2019, Sec. 5.2) and Ghil and Lucarini (2020, Sec. IV.E). Thus the response to the latter — e.g., the decay time of an exogenous shock's effect on the system — can be determined from the system's typical lag-autocorrelation time.

---

[4]https://www.nber.org/cycles/cyclesmain.html, consulted on April 6, 2020; pdf version dated April 23, 2012

Section 5 concluded by reviewing a query into the existence of a worldwide synchronization of economic activity, resulting in a positive conclusion, cf. Fig. 9 herein. Synchronization, though, depends sensitively on the coupling parameters between chaotic oscillators (e.g., Colon and Ghil, 2017; Duane et al., 2017, and references therein). So far, the main concerns of IAM-based investigations of overall, world-wide economic effects of climate change, as well as those of specific studies of

5 more localized economic effects of extreme climatic and other natural events have focused on physical losses of economic productivity. Sensitive dependence of synchronization on parameter values suggests that more subtle, but still calamitous productivity losses could arise from climatically driven changes in the world economic activity's degree of synchronization.

Finally, we note that significant steps have been taken of late to achieve insights into an even broader system, beyond climate and the economy, to encompass sociological aspects of climate change impacts as well (e.g., Motesharrei et al., 2014, 2016).

Useful pointers in the direction of dynamic modeling of sociological problems can be found in the fairly obvious analogy between the latter and ecological ones, as noted by Samuelson (1971), May et al. (2008) and Colon et al. (2015), among others.

To cover these developments would require expanding the present review much further, which is not in the cards at this time. Still, the answer to Problem 10 of Ghil (2001), namely

"Can we achieve enlightened climate control of our planet by the end of the century?"

does require a complete understanding of the behavior of such a coupled socio-economic–physical–biological system, along with a deeper understanding of who "we" are and who exercises the control.

## Appendix A. Local variance and the FDT

Given the importance of the FDT conjecture for studying the economic system, we provide herein a quick introduction to the local variance concept used in Sec. 5.1 and illustrated in Fig. 8(c). Please see Ghil et al. (2002) and Groth et al. (2015a) for

the precise definitions and equations used in the M-SSA methodology, and Groth et al. (2015b) for the details of the statistical significance tests applied to the BEA dataset.

Plaut and Vautard (1994) introduced the concept of *local variance fraction* $V_{\mathcal{K}}(t)$,

$$V_{\mathcal{K}}(t) = \frac{\sum_{k \in \mathcal{K}} A_k(t)^2}{\sum_{k=1}^{DM} A_k(t)^2}, \tag{5}$$

which quantifies the fraction of the total variance that is described by a subset $\mathcal{K}$ of SSA principal components (temporal PCs

or T-PCs) in a sliding window of length $M = 24$ quarters. The T-PCs here are considered as centered, i.e. starting at $M/2$ and ending at $N - M/2$, and $D = 9$ is the dimension of the phase space into which the macroeconomic indicators are embedded, cf. Groth et al. (2015b, Secs. 2.2 and 2.3). Here $N$ is the total length of the time series, with $N = 52$ years $\times 4$ quarters $= 208$ data points for each time series.

The leading oscillatory mode plotted in Fig. 8(b) corresponds to $\mathcal{K}_0 = \{k = 1, 2\}$ and it is considered as the signal, while the

30 complementary set $\mathcal{K}_1 = \{k = 3, \ldots, DM = 216\}$ is identified with the fluctuations whose evolving variance we wish to track. More precisely, a total of $\mathcal{K}' = \{1 \leq k \leq 150\}$ eigenvalues in the M-SSA decomposition capture 99 % of the BEA dataset's

total variance. Hence, the local variance plotted in Fig. 8(c) corresponds to $\mathcal{K}'' = \{3 \leq k \leq 150\}$.

*Acknowledgements.* It is a pleasure to thank Daniel Schertzer for the invitation to write an update of the Ghil (2001) paper for the *Centennial Issue on Nonlinear Geophysics: Accomplishments of the Past, Challenges of the Future* of the *Nonlinear Processes in Geophysics*. Sections 2 and 3 of this review have benefited from interaction with many climate colleagues over the years; quite a few of them have been acknowledged recently in Ghil (2019) and in Ghil and Lucarini (2020). In connection with Secs. 4 and 5, it is a great pleasure to thank M. D. Barnett, E. Biffis, W. A. Brock, E. Chavez, C. Chiarella, D. Claessen (RIP), C. Colon, B. Coluzzi, P. Dumas, A. Groth, S. Hallegatte, L. P. Hansen, J.-C. Hourcade, M. Nikolaidi, M. Sadler, L. Sela, P. Terna, G. Vivaldo and G. Weisbuch for all they taught me about economics and its modeling. Figure 7 was produced by Andreas Groth, to whom many thanks. I am deeply indebted to William A. Brock, Célian Colon, Maria Nikolaidi and Pietro Terna for enlightening comments on a draft of this paper, especially on its Secs. 4 and 5. Two anonymous reviewers and Valerio Lucarini have made highly constructive and helpful suggestions that have further improved the presentation, while Stéphane Vannitsem's careful reading helped me identify and fix a typo in Eq. (4). This paper is TiPES contribution #19; the TiPES (Tipping Points in the Earth System) project has received funding from the European Union's Horizon 2020 research and innovation program under Grant Agreement No. 820970. Work on this paper has also been supported by the EIT Climate-KIC; EIT Climate-KIC is supported by the European Institute of Innovation & Technology (EIT), a body of the European Union.

The author declares that he has no conflict of interest.

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
