# Peer review of "Hilbert problems for the climate sciences in the 21st century – 20 years later"

_Nonlinear Processes in Geophysics, 2020_

## Referee Comment (RC1) · Anonymous Referee #1 · 9 Jun 2020

The author updates his discusses of ten major problems in climate science outlined in Ghil (2001). This interesting paper is not a standard review of a certain area of climate science, but more of a career overview and perspective of the author, spanning multiple research areas and different intersections. The problems discussed start with atmospheric and oceanic variability modes, and continue to economic, and coupled climate-economic problems. Throughout, the author takes dynamical system theory as the main tool and applies it to this broad range of problems. I enjoyed very much reading the paper, found some parts to be surprising and other parts led me to search for subjects or tools that I was not aware of. It seems that this would be the exact purpose of such a paper - expanding the areas of interest of the readers that are likely to come from one specific specialization area, and I think the paper achieves that goal

very nicely. I recommend publication basically as is. A couple of minor comments: (1) "climate control" made me think of geoengineering, and I was glad to see that was not the intention. Perhaps this can be made more explicit where necessary (or, even better, perhaps the terminology could be changed?). (2) These seem to be not Hilbert problems for the Geosciences, but more like for climate sciences, which is broad enough as is, of course. Perhaps the author wishes to consider renaming the manuscript or briefly putting the selected problems in the perspective of the broader Geosciences that might require a very different set of problems.

---

## Author Comment (AC1) · 9 Jun 2020

Thank you for the encouraging comment. Except that the career is far from over ;-)

---

## Referee Comment (RC3) · Anonymous Referee #2 · 20 Jun 2020

This paper is essentially an essay, based on an earlier paper written by the same author at the turn of the millennium which posed a set of ten pivotal "big picture" problems facing the geosciences at the time, particularly in relation to climate dynamics, prediction and the possibility of the eventual "enlightened" control of our planetary environment. The vision presented to the reader is on an immensely broad canvas and presents what is clearly an authoritative though personal view of progress (and none the worse for this!), from the perspective of a scientist/theoretician who has been a prolific and influential proponent of the dynamical systems approach towards understanding and modelling the behaviour of complex systems.

The author wisely restricts his "20 years later" update to reviewing just a subset of his Hilbert problems, focusing on both some recent advances (a) in modeling aspects of

the physical climate system (notably on low frequency variability on timescales ranging from the inter-annual to the multi-decadal) in the atmosphere and oceans, and also (b) on the formulation of approaches to modelling the dynamics of global and regional economies and their possible interactions with and responses to variations in the climate. In (a), it is evident that advances in both analysis methods (such as Ghil's own approach to multivariate singular systems analysis) and modelling approaches based on non-autonomous and random dynamical systems have led to new insights into systems as complex as the coupled atmosphere-ocean climate system. In particular, that they can develop much more complicated cyclic or chaotic behaviour than simple, steady equilibria with timescales that are much longer than those of imposed forcings, through bifurcations between pullback attractors - including the intriguing posslibities of more than one such attractor coexisting at the same point in parameter space. Such possibilities have important lessons and implications for scientists modelling these systems and using them to predict future climate responses to changes in forcing.

Particularly intriguing (certainly for this physical scientist!) are the author's recent forays, reviewed in (b), into applying his dynamical systems approaches (alongside others) to similar kinds of problem facing economists in trying to capture the complex dynamics of large scale economies and understand how they may respond to external changes in the environment, especially involving extreme, high impact events. From his perspective as an expert in dynamical systems, Prof. Ghil takes a fascinatingly critical view of how economists have approached the problem of formulating macroeconomic models over time. In particular, he notes some parallels in how ideas have evolved in economics with the world of climate dynamics, in which early ideas focused on predicting and understanding evolution towards simple equilibria but more recent work has moved on to explore non-equilibrium (cyclic and/or chaotic) behaviours, both exogenous and endogenous, which appear to show greater promise in emulating how real economies actually behave.

The overall result is a fascinating, concise (though in places perhaps a little too connone
cise?) and quite readable synthesis which indicates some real progress in pulling together these diverse and important topics in both geosciences and the social sciences. The article is well referenced, and would certainly be expected to stimulate many readers to follow up some of these ideas in the cited literature. My only gripes with the paper in its present form are summarised in the short list of questions and clarifications below. But hopefully the author will take these on board in revising his paper for final publication.

Specific points:

P.8 lines 1-2. The reference here to "an important step in achieving greater rigor in this field is a greater reliance on the counterfactual theory for necessary and sufficient causation (Pearl, 2009) in the attribution of such extreme events" reads somewhat strangely. How can a "counterfactual theory" be rigorous (let alone correct?)? Perhaps "counter-intuitive" is intended here, or I may have completely misunderstood what is intended? Otherwise some additional explanation on what is meant (and what the "theory for necessary and sufficient causation " actually is) would be welcome - especially since the book cited is not likely to be widely accessible to many readers.

Figure 3 and associated text. Some further explanation of both the context of this figure (perhaps specifying in more detail the DDE being solved and the relevant parameters?) and what the (unlabelled) z axis represents would be very welcome here.

P.3 introduction: The end of this section seems to suggest that we will learn about progress on Ghil's problem (10) "Can we achieve enlightened climate control of our planet by the end of the century?" in later sections. But the actual discussion in sections 4 and 5 don't really address this question, at least not directly. Are we anywhere near being able to answer it - perhaps with emphasis not only on the technical feasibility but also on the "enlightened" nature of the policies underpinning such control? Some more discussion in Section 6 might bring this to a more satisfying conclusion?

P. 10 line 22 "heterogeities" [typo]

P.19 line 19. "Atmosphetic" [typo]

---

## Author Comment (AC2) · 24 Jun 2020

Thank you very much for this thoughtful, in-depth and encouraging review.

The points raised at the end are well taken and will be obviously addressed carefully and thoroughly in the revised version of the paper.

At this point, I will merely try to elucidate the misunderstanding about the "counterfactual theory for necessary and sufficient causation" appearing on p. 8, ll. 1-2 of the paper's published preprint under discussion. This theory, as presented in particular in the now classical book of Judea Pearl on "Causality: Models, Reasoning and Inference" (CUP, 2nd edition, 2009), brought the author the ACM's 2011 Turing Award, known as the Nobel Prize of the Computer Sciences. It was obviously an error to

assume that, as such, the theory would be known to the NPG readership at large.

The revised version of the paper will expand on the theory – which is at the same considered to be an important contribution to the Philosophy of Science, Statistical Theory and Methods, Statistics and Probability, and Philosophy per se – and explain in which way its requirements of necessity and sufficiency in causation criteria are distinct from the usual causation criteria applied in the climate sciences for detection and attribution of extreme events. The review paper below is open access and the reference will also be added to the revised version of npg-2020-13.

Reference

Pearl, J.: Causal inference in statistics: An overview. Statistics Surveys, 3, 96–146, doi: 10.1214/09-SS057, 2009.

---

## Short Comment (SC1) · 1 Jul 2020

A really good paper, very instructive and very readable. It is extremely useful to have a recap of scientific progress in some extremely important areas of long-term research in geosciences, and the author's personal perspective is very intriguing.

I have especially enjoyed Sects 4.2 and 5, as these are research areas I am way less familiar with. My clear impression is that they contain a lot of interesting and innovative ideas. I always had the vague intuition that delays in the response of the economic system to perturbations are associated with some form of irreversibility/nonequilibrium, and it was very satisfying to read the extremely insightful discussion and conjectures b the author.

[Figure]

Specifically, of great interest is the discussion of the FDT in the context of economics. I would recommend the author to add an explicit statement like the one at page 20 in the conclusions (".. To wit, internal endogenous fluctuations are likely to change in variance with the phase of the business cycle in the same way as the exogenously driven ones, i.e., larger "volatility" can be expected during expansions than during contractions of the economy."..) already at page 16, because the link between amplitude of the fluctuations and local sensitivity is not spelled out in a sufficiently explicit way, in my personal opinion. Maybe the author should expand a bit.

A small typo: Christian Huyghens -> Christiaan Huygens

---

## Author Comment (AC3) · 1 Jul 2020

Thank you for the overall assessment of the paper and, more specifically, for the suggestion on expanding the text concerning applications of the fluctuation-dissipation theorem (FDT) in macroeconomics. The latter will be pursued in the revised version as suggested. Thank you also for the correct spelling of the pioneering Dutch physicist's name: I was citing from memory and was, obviously, wrong for once,
* * *

---

## Author Response (AR1)

Author's Response to Reviews of ms. npg-2020-13,
*Hilbert problems for the geosciences in the 21st century – 20 years later*
by Michael Ghil

Three *interactive comments* were posted during the *Discussion* of this ms., by anonymous Referees #1 and #2, and by Valerio Lucarini. All three were strongly encouraging and I am grateful for the thoughtful comments and suggestions made. As indicated in my online responses to all three, the comments and suggestions have been fully taken into account in the revised version submitted herewith.

Below are the individual comments, in **black and Roman font**, and the responses*, in blue and italics*, along with the pages and lines of new text in the compiled latexdiff version included as part of this Response.

**Referee #1**

The author updates his discusses [*sic*] of ten major problems in climate science outlined in Ghil (2001). This interesting paper is not a standard review of a certain area of climate science, but more of a career overview and perspective of the author, spanning multiple research areas and different intersections. The problems discussed start with atmospheric and oceanic variability modes, and continue to economic, and coupled climate-economic problems. Throughout, the author takes dynamical system theory as the main tool and applies it to this broad range of problems. I enjoyed very much reading the paper, found some parts to be surprising and other parts led me to search for subjects or tools that I was not aware of. It seems that this would be the exact purpose of such a paper - expanding the areas of interest of the readers that are likely to come from one specific specialization area, and I think the paper achieves that goal very nicely. I recommend publication basically as is.

*Thank you very much for this fine summary and for the encouragement.*

A couple of minor comments: (1) "climate control" made me think of geoengineering, and I was glad to see that was not the intention. Perhaps this can be made more explicit where necessary (or, even better, perhaps the terminology could be changed?).

*The correct intent has now been explained in the first paragraph of Sec. 4.1, ll. 5–10 on p. 8 of the revision's latexdiff version below. The paragraph specifically excludes "crude geoengineering proposals" from "enlightened climate control." Thank you for suggesting the clarification.*

(2) These seem to be not Hilbert problems for the Geosciences, but more like for climate sciences, which is broad enough as is, of course. Perhaps the author wishes to consider renaming the manuscript or briefly putting the selected problems in the perspective of the broader Geosciences that might require a very different set of problems.

*The title of the paper has been changed as suggested.*

**Referee #2**

This paper is essentially an essay, based on an earlier paper written by the same author at the turn of the millennium which posed a set of ten pivotal "big picture" problems facing the geosciences at the time, particularly in relation to climate dynamics, prediction and the possibility of the eventual "enlightened" control of our planetary environment. The vision presented to the reader is on an immensely broad canvas and presents what is clearly an authoritative though personal view of progress (and none the worse for this!), from the perspective of a scientist/theoretician who has been a prolific and influential proponent of the dynamical systems approach towards understanding and modelling the behaviour of complex systems.

The author wisely restricts his "20 years later" update to reviewing just a subset of his Hilbert problems, focusing on both some recent advances (a) in modeling aspects of the physical climate system (notably on low frequency variability on timescales ranging from the inter-annual to the multi-decadal) in the atmosphere and oceans, and also (b) on the formulation of approaches to modelling the dynamics of global and regional economies and their possible interactions with and responses to variations in the climate. In (a), it is evident that advances in both analysis methods (such as Ghil's own approach to multivariate singular systems analysis) and modelling approaches based on non-autonomous and random dynamical systems have led to new insights into systems as complex as the coupled atmosphere-ocean climate system. In particular, that they can develop much more complicated cyclic or chaotic behaviour than simple, steady equilibria with timescales that are much longer than those of imposed forcings, through bifurcations between pullback attractors - including the intriguing posslibities of more than one such attractor coexisting at the same point in parameter space. Such possibilities have important lessons and implications for scientists modelling these systems and using them to predict future climate responses to changes in forcing.

Particularly intriguing (certainly for this physical scientist!) are the author's recent forays, reviewed in (b), into applying his dynamical systems approaches (alongside others) to similar kinds of problem facing economists in trying to capture the complex dynamics of large scale economies and understand how they may respond to external changes in the environment, especially involving extreme, high impact events. From his perspective as an expert in dynamical systems, Prof. Ghil takes a fascinatingly critical view of how economists have approached the problem of formulating macroeconomic models over time. In particular, he notes some parallels in how ideas have evolved in economics with the world of climate dynamics, in which early ideas focused on predicting and understanding evolution towards simple equilibria but more recent work has moved on to explore non-equilibrium (cyclic and/or chaotic) behaviours, both exogenous and endogenous, which appear to show greater promise in emulating how real economies actually behave.

The overall result is a fascinating, concise (though in places perhaps a little too concise?) and quite readable synthesis which indicates some real progress in pulling together these diverse and important topics in both geosciences and the social sciences. The article is well referenced, and would certainly be expected to stimulate many readers to follow up some of these ideas in the cited literature. My only gripes with the paper in its present form are summarised in the short list of questions and clarifications below. But hopefully the author will take these on board in revising his paper for final publication.

*Thank you very much for this thorough and extensive summary and for the encouragement. The slight criticism of the paper being "*perhaps a little too concise" *has led to the points raised by the referee below being addressed in hopefully a satisfactory and not too concise manner.*

**Specific points:**

P.8 lines 1-2. The reference here to "an important step in achieving greater rigor in this field is a greater reliance on the counterfactual theory for necessary and sufficient causation (Pearl, 2009) in the attribution of such extreme events" reads somewhat strangely. How can a "counterfactual theory" be rigorous (let alone correct?)? Perhaps "counter-intuitive" is intended here, or I may have completely misunderstood what is intended? Otherwise some additional explanation on what is meant (and what the "theory for necessary and sufficient causation " actually is) would be welcome - especially since the book cited is not likely to be widely accessible to many readers.

*Thank you for your interest in the* "counterfactual theory for necessary and sufficient causation (Pearl, 2009)." *This interest has led to introducing a whole new subsection on "Detection and Attribution (D&A) studies," p. 8, l. 15 to p.10, l. 5 of the latexdiff version below. The criticism of the widely lauded book by J. Pearl (2009a) not being easily available led to adding a citation to an open-access review paper by the same author (Pearl, 2009b).*

Figure 3 and associated text. Some further explanation of both the context of this figure (perhaps specifying in more detail the DDE being solved and the relevant parameters?) and what the (unlabelled) z axis represents would be very welcome here.

*Thank you for your interest in this figure and the associated text. This interest has led to another new subsection of the paper, titled "Beyond equilibrium climate sensitivity," p. 10, ll. 6–31. The caption of the figure has also been modified accordingly.*

P.3 introduction: The end of this section seems to suggest that we will learn about progress on Ghil's problem (10) "Can we achieve enlightened climate control of our planet by the end of the century?" in later sections. But the actual discussion in sections 4 and 5 don't really address this question, at least not directly. Are we anywhere near being able to answer it - perhaps with emphasis not only on the technical feasibility but also on the "enlightened" nature of the policies underpinning such control? Some more discussion in Section 6 might bring this to a more satisfying conclusion?

*Thank you for raising this truly crucial point. It is hoped that the current version's final lines will satisfy you. They are cited here in full:*

*"Finally, we note that significant steps have been taken of late to achieve insights into an even broader system, beyond climate and the economy, to encompass sociological aspects of climate change impacts as well (e.g., Motesharrei et al., 2014, 2016). Useful pointers in the direction of dynamic modeling of sociological problems can be found in the fairly obvious analogy between the latter and ecological ones, as noted by Samuelson (1971), May et al. (2008) and Colon et al. (2015}, among others.*

*To cover these developments would require expanding the present review much further, which is not in the cards at this time. Still, the answer to Problem 10 of Ghil (2001}, namely ``Can we achieve enlightened climate control of our planet by the end of the century?'' does require a complete understanding of the behavior of such a coupled socio-economic-- physical--biological system, along with a deeper understanding of who ``we'' are and who exercises the control."*

P. 10 line 22 "heterogeities" [typo] & P. 19 line 19. "Atmosphetic" [typo]

*Fixed, thank you.*

**Referee #3, Valerio Lucarini**

A really good paper, very instructive and very readable. It is extremely useful to have a recap of scientific progress in some extremely important areas of long-term research in geosciences, and the author's personal perspective is very intriguing. I have especially enjoyed Sects 4.2 and 5, as these are research areas I am way less familiar with. My clear impression is that they contain a lot of interesting and innovative ideas. I always had the vague intuition that delays in the response of the economic system to perturbations are associated with some form of irreversibility/nonequilibrium, and it was very satisfying to read the extremely insightful discussion and conjectures by the author.

Specifically, of great interest is the discussion of the FDT in the context of economics. I would recommend the author to add an explicit statement like the one at page 20 in the conclusions (".. To wit, internal endogenous fluctuations are likely to change in variance with the phase of the business cycle in the same way as the exogenously driven ones, i.e., larger "volatility" can be expected during expansions than during contractions of the economy."..) already at page 16, because the link between amplitude of the fluctuations and local sensitivity is not spelled out in a sufficiently explicit way, in my personal opinion. Maybe the author should expand a bit.

*Thank you very much for this insightful summary and especially for your interest in "*Sects 4.2 and 5" *and* "the discussion of the FDT in the context of economics," *which are the key novelties in this review-&-perspective paper. The suggested expansion occurred in the 2nd paragraph of*

*Sec. 5.2, p. 19, ll. 17–21 of the attached latexdiff version of the revised ms., as well as in the Concluding remarks, p. 23, ll. 24–29, and in the following paragraph, p. 23, l. 34–p. 24, l. 2.*

A small typo: Christian Huyghens -> Christiaan Huygens

*Fixed, thank you.*

*A number of other, more minor changes and additions have been made. They all appear in the attached latexdiff version of the revised ms.*

[revised manuscript text omitted]